# Yarn: Adding Meaning to Shared Personal Data through Structured Storytelling

Daniel A. Epstein*

Informatics

University of California, Irvine

Mira Dontcheva†

Adobe Research

James Fogarty‡

Computer Science & Engineering

University of Washington

Sean A. Munson§

Human Centered Design & Engineering

University of Washington

## ABSTRACT

People often do not receive the reactions they desire when they use social networking sites to share data collected through personal tracking tools like Fitbit, Strava, and Swarm. Although some people have found success sharing with close connections or in finding online communities, most audiences express limited interest and rarely respond. We report on findings from a human-centered design process undertaken to examine how tracking tools can better support people in telling their story using their data. 23 formative interviews contribute design goals for telling stories of accomplishment, including a need to include relevant data. We implement these goals in Yarn, a mobile app that offers structure for telling stories of accomplishment around training for running races and completing Do-It-Yourself projects. 21 participants used Yarn for 4 weeks across two studies. Although Yarn's structure led some participants to include more data or explanation in the moments they created, many felt like the structure prevented them from telling their stories in the way they desired. In light of participant use, we discuss additional challenges to using personal data to inform and target an interested audience.

**Keywords**: Personal data; personal informatics; sharing; storytelling; authoring; self-tracking; social networking sites.

**Index Terms**: Human-centered computing → Collaborative and social computing; social media; • Human-centered computing → Ubiquitous and mobile computing → smartphones

## 1 INTRODUCTION

Devices and apps that track personal data have become increasingly pervasive, including Fitbit for activity, Mint for finances, and Swarm for location. These apps often include social features that allow people to share with others who collect the same type of data (e.g., leaderboards on Fitbit, Spotify's friends feed, non-place check-ins [9]) or from a broader audience of friends and family (e.g., over a social network like Facebook or Twitter or over SMS, [10,43,44]). Prior work has suggested that sharing experiences, when designed well, can allow the person sharing to celebrate achievements [64], get advice [38,42,55], and be held accountable to their goals [7,58]. Audiences can learn more about the person sharing [3,36] and feel more connected to them [2,25].

For some people, digitally sharing personal data has led to positive experiences, such as support, accountability, and connectedness when sharing among small groups of close connections [25,60] or in finding communities of strangers via hashtags [7]. But more often, potential sharers express concerns about posting content that audiences may think of as boring [15,43,46]. Audiences similarly express limited interest and rarely respond to shared content [16,37,43]. Shared content often appears system-generated, such as updates or numerical summaries (e.g., "*Julia is listening to U2*", "*Eliott took 8,423 steps today*"), which can come across as impersonal and disconnected from the story that motivated a person to track [16]. In a review of people's social needs for personal tracking, Kersten-van Dijk & Ijsselsteijn suggest "*to move beyond impersonal, standard messages to fostering true connections between self-trackers and their various audiences, [designs] need to support those users in telling their story and sharing experiences with their data, their way*" [28].

This call by Kersten-van Dijk and Ijsselsteijn motivates our research question: How can tracking tools better support people in telling their story and sharing their experiences with data, their way? Within this overall question, we also ask three sub-questions: (1) What kinds of stories do people feel personal data is well-suited to help tell?, (2) How can personal data be integrated in ways which reflect the person's goal for sharing and feel personal?, and (3) Can a design help people create content which aligns with their goals and leads to support from others?

We undertook a human-centered design process to answer these questions. We conducted formative interviews with 23 people. These interviews informed key design principles, including a need to align content with sharing motivation. We prototyped and iterated on the design of the Yarn mobile app. Compared to common social media platforms, Yarn's key novel design features included telling persistent stories by linking shared moments, dividing stories by goals in separate chronological feeds, and visual templates which reflected a person's motivation for sharing. We deployed Yarn to understand whether Yarn's strategies achieved the design principles through two field evaluations with 21 total people who used Yarn for 4 weeks.

Though Yarn's design strategies were a modest success, participant's experiences and perspectives also offer further guidance for how future tracking tools might better support people in sharing stories as they wish to. Through our human-centered design process, we contribute:

- An understanding of the types of stories people want to use personal data to tell. We focus on people's use of tracked data to help share stories of accomplishment which unfold over weeks or months, such as completing home Do-It-Yourself Projects or training for running races.
- Four design goals for telling their stories of accomplishment. (1) share throughout a story's process, (2) present stories as

* e-mail: epstein@ics.uci.edu
† e-mail: mirad@adobe.com
‡ e-mail: jfogarty@cs.washington.edu
§ e-mail: smunson@uw.edu

chronological chapters, (3) include relevant data, and (4) emphasize and explain important moments.

- Five design strategies implemented in a mobile app, Yarn, which embody the four design ideas: (1) optional and automatic data entry, (2) visual templates, (3) description suggestions, (4) automatic inference of importance, and (5) as-desired sharing.

- Evaluation of these design strategies. Yarn's structure led some participants to include more data or explanation in the moments they created, while others ignored the guidance entirely. Many participants felt that the structure Yarn imposed prevented them from telling their stories, their way. Most participants desired more support and advice than they received.

- Discussion of the outcomes of our human-centered design process. Given the mixed reaction to Yarn's design strategies, storytelling tools should support further documentation for personal purposes alongside sharing to engage peers and close ties and offer more flexibility in how moments are presented.

## 2 BACKGROUND

Our design process was motivated by prior literature describing people's motivations for sharing personal data and strategies used in digital tools for sharing and storytelling.

### 2.1 People's Motivations for Sharing Personal Data

In HCI research, personal data has been used to refer to digital items people collect and retain about themselves (e.g., personal photos or videos for reminiscence [23], self-tracked data for reflection [33,34]), items on digital devices owned by a person (e.g., work files, emails [61]), or anything digital generated by the person (e.g., social media content [20,26]). There is no widely agreed upon definition of personal data [62]. We define personal data as *anything people intentionally collect and retain about themselves for later self- or collaborative-reflection or reminiscence*. This definition relates to data collected from personal informatics systems, defined by Li et al. as systems which "*help people collect personally relevant information for the purpose of self-reflection and gaining self-knowledge*" [33]. Personal informatics shares common motivation with the Quantified Self movement, which emphasizes "*self-knowledge through numbers*" [5,33]. But personal informatics data or personal data need not be automatically collected, numeric, or even digital. Journals promote similar self-reflection goals. Many digital journals integrate open-ended text diary features with numeric data such as activity and weather with uploaded media [14]. For our purpose of designing a tool for digitally sharing personal data, we leverage personal data that is digitally-produced (e.g., photos or videos, GPS traces) or digitally-logged (e.g., a journal app).

People often share to get or give *recommendations or advice* [42]. Exchange of advice is often a primary goal of peer support communities, such as for health [38], overcoming cancer [55], and personal finances [56]. People often share personal data to give context (e.g., biometrics associated with progress, photos of weight loss) [27,55]. People describe the practices which have worked for them to help others achieve similar goals.

People also share their personal data for *emotional support, motivation, or accountability* [58]. For example, some people follow hashtags on Instagram to find and receive support from others with similar health goals [7]. Some then use those hashtags in their own posts to contribute to the community, and they felt guilty for letting down their followers whenever they do not post something they ate or post something unhealthy.

Finally, others use personal data to *share an achievement* they are proud of [64]. In these cases, personal data can serve as a record or better explain the achievement, such as a location trace from a race or photos from an artistic project [17]. Some people do this to curate an impression of themselves (e.g., as an adventure-seeking person, as a healthy person) [24,64]. Others share to become closer to their sharing audience [63].

Commercial and research apps use a variety of features to support these sharing goals. Some seek to create communities within the app or platform for sharing ideas and opinions [29,47]. Apps that track physical activity often promote interpersonal competition, such as leaderboards [8,25,35] or daily challenges [18]. When people have a common context around shared challenge, goals, or accomplishments, sharing data is often enough to facilitate an engaging experience. For example, a running community understands the accomplishment of a long training run, and a diet community understands the struggle against temptation.

Some apps for collecting personal data facilitate reaching people who do not use the app, such as through broad social networking platforms like Facebook and Twitter (e.g., [43,44]) or through direct communication via SMS or Email (e.g., [37,43]). Commercially, running apps like Nike+, Strava, and RunKeeper enable sharing routes ran or photos taken on the run annotated with information about distance and pace. This broader sharing allows people to share with and receive feedback from friends and family, people whose support can be particularly meaningful [46].

Unfortunately, apps which share to these broader social networking platforms do not help people create content that conveys *why* they are sharing their personal data, and thus audiences must infer the sharer's motivations [16,37]. As a result, people often do not receive the responses the seek, or any response at all [16,43]. Many apps automatically push data to these social platforms when it is collected (e.g., when food is logged in MyFitnessPal or a run in RunKeeper), removing opportunity for explanation [6,16]. Though some apps support adding text or photos, these fields are rarely used. When people use them, they often describe what they did rather than why it is important [16].

### 2.2 Sharing and Storytelling Strategies in Digital Tools

Designing technology to support storytelling has a rich history in HCI, including digital systems for advocacy and social movements [12,40], digital cultural probes which help people describe their lives [21], and data-driven approaches to storytelling [49,53]. We focus our review on the strategies used by designs which integrate personal data into stories.

One common design strategy is *a structured authoring experience* which integrates how experts organize their stories. For example, the Motif system surfaces types of video shots that will create a good narrative structure, helping people capture and assemble video stories in the moment [30]. DataSelfie encodes and maps personal data to enable people to choose how they want to visually represent their identities [31]. Having to physicalize personal data can also structure creation [59]. Other systems aid people in organizing previously taken photos and videos into a story. The iTell and Storied Navigation systems use prompts to help people effectively brainstorm what they want to highlight and organize any associated data [32,54]. Other systems have instead helped people search within the photos, videos, and location data they collect for relevant or memorable moments. The classic MyLifeBits system included a map and calendar for browsing location-tagged photos [22]. The Raconteur conversational agent mined text conversations with a friend to identify what photos or videos may be appropriate to share [4].

Other designs help people *document the important moments in their progress*, aligning with Rooksby et al.'s documentary-driven

tracking style [50]. A record of important moments can help people tell the story of their progress after completion or can help people get advice along the way. For example, the Mosaic system provides a structured process for people to share creative works-in-progress for early feedback [29]. Spyn similarly supports documenting the experience of knitting through photos and location data to help people demonstrate their progress [51]. Smart journals, or digital personal diaries, use a range of media to help people author their own histories [14]. Though journaled data is typically collected for private consumption, Elsden et al. find that the data can serve as a talking point with others when the authored content prompts curiosity [13].

Prominent social platforms including Instagram, Snapchat, and Facebook include story features which provide flexible tools for annotating and sharing data ephemerally. People tend not to worry whether audiences will find shared stories interesting because they are only visible for a day and viewing is voluntary [39]. In a larger storytelling context, when using these sharing features to share a story of accomplishment, the ephemerality can result in audiences missing important moments and the cumulative effort undertaken by the sharer. However, ephemerality may also align well with the contributions of minor moments to a story.

## 3 FORMATIVE INTERVIEW INSIGHTS

To answer our first research question, *what kinds of stories do people feel personal data is well-suited to help tell?*, we conducted two sets of formative interviews. We recruited both sets of participants through posts to community Facebook groups and messages to university email lists. There was no overlap in participants between the two sets of interviews.

In the first set, we sought to understand what sorts of stories people wanted to use personal data to tell. We interviewed 16 people in their homes (9 identified as female, 7 as male, ages 25-40). Three participants were currently students, four were in technology-related fields. The remainder ranged from teachers to marketing consultants to customer service representatives.

We asked interviewees to describe the personal data they collect and to brainstorm stories they were interested in telling through that data. They then completed a design activity where they sketched on paper how they would want to tell that story. We then asked questions to better understand the designs they sketched. The first author conducted all of the interviews, taking field notes and recording audio, sending the audio to an external service for transcription.

The stories participants imagined led us to narrow our design exploration to stories of accomplishment, specifically home Do-It-Yourself (DIY) projects and training for running races. With a story and two domains selected, we sought to better understand how people want to tell those specific types of stories. We interviewed 7 people, 3 who were interested in telling running stories (2 F, 1 M) and 4 who were interested in telling DIY stories (3 F, 1 M), ages 27-35. Occupations ranged from sales to stay-at-home parents to public relations. These participants had all either recently completed or were in the middle of the accomplishment. One race participant was training for their third half marathon, while the other two were training for their first full marathon. DIY projects included a table, a dollhouse, a bookshelf, and a living room renovation. We followed a similar protocol to the previous interviews, framing the interview around their accomplishments and asking participants to sketch how they would want to present that story.

We used qualitative data analysis [41], to analyze both sets of interviews through bottom-up thematic analysis through open coding [57]. The first author then re-coded the interviews, reviewed the designs participants sketched, and wrote memos summarizing themes, discussing and refining these themes with other members of the research team. We quote participants as I1-23. The first 16 participants were in the first round of interviews (e.g., I1-16). The remainder were in the second round, where we include a superscript for type of story (e.g., I17run, I18diy).

We found four trends in how participants wanted to tell their stories. Participants wanted to:

(1) **Share throughout a story's process**. Participants expected that some audiences would appreciate seeing intermediate progress, but others would only want to see the final result. For example, I21diy imagined how she might want to share her dollhouse project: "*I only shared the process photos with my husband and my mom. I feel like the process isn't probably that interesting to other people. Maybe it would be, but from what I've talked with people they like to see the final thing.*"

(2) **Present their stories chronologically**. All but one participant wanted to present their story chronologically to others, reflecting how they experienced the story and similar to techniques used in prior storytelling designs (e.g., [30,54]). Participants drew timelines, calendars, and infinite scrolling to represent an ordered view. These activity streams looked much like social media feeds, but focusing on one person's data allowed for audiences and posters to understand how a story has progressed, developed, and evolved.

(3) **Include a range of data types**. Participants had diverse opinions about what kind of data they wanted to include in their stories including photos, text descriptions, distances, expenses, weather, and locations. Some data would be required, but others optional. For example, I23run wanted to take a picture of each run, with other data in a box overlaying the image: "*then, this box would be data. I would want to share the distance, and then optionally, duration, pace.*"

(4) **Emphasize and explain important moments**. People are often concerned that sharing trivial accomplishments or progress will bore their audience [15,16,43], a concern shared by our participants. Participants wanted their minor moments, such as training runs or repetitive DIY tasks, contributed to their overall story. For example, I17run felt that "*the most important aspect to convey to people who are not runners is the level of commitment that this takes… Totals are cool to me, how many miles did you run for this marathon, how long did it take.*"

## 4 YARN'S STORYTELLING APPROACH

To answer our second research question, *how can personal data be integrated in ways which reflect the person's goal for sharing and feel personal?*, we designed and developed the Yarn mobile app to incorporate the trends we observed in our formative interviews. We iteratively designed Yarn with low-fidelity and mid-fidelity mockups, getting feedback from other members of the research team throughout the process. We implemented Yarn for iOS.

We iteratively designed Yarn to satisfy the design goals identified in our formative interviews. Yarn primarily consists of five pages. (1) The home screen lists all stories that the author is working on. After clicking on a story, the author sees (2) a chronological feed of all chapters which make up that story. Creating a new chapter follows a similar model to Instagram's flow of selecting pictures and an appropriate filter. The author first clicks a "+" icon to bring up (3) the data selection screen, where they then choose what data to log for the chapter. They then access (4) the template selection screen, where they choose how they want to present the data they logged. The design iteration introduced a (5) community screen, where authors can see the posts made by other people using Yarn. The supplemental video demonstrates Yarn's application flow, creating a chapter and sharing it to a social networking site.

Many prior research and commercial tools for sharing personal data automatically produce a message or post to share based on tracked or journaled data [16]. When configured, some automatically share whenever new data is tracked or journaled. Yarn instead supports a person in authoring their own content, integrating that tracked our journaled data into the content they create. Although Yarn suggests information to include and presentation formats, the author ultimately decides what content to generate and whether to share it.

Yarn implements five design strategies (DSs) informed by the trends surfaced in our formative interviews.

## 4.1  Optional and Automatic Data Entry (DS1)

The data selection screen (Figure 1) includes five categories of data an author can log. We categorized the data type recommendations given by our formative interviewees according to how the data is typically presented (e.g., as an image, as a number, as text):

- **Visual data**: a visual indication of progress toward the accomplishment. People often find visual data such as photos more interesting than just text and numbers [16]. In DIY projects, photos and videos comprised the visual data. Race training stories also included route maps as visual data.
- **Numeric data**: any numeric measurements of progress toward the accomplishment. We selected *primary* and *secondary* fields based on how formative interviewees imagined tracking their progress. We selected time worked and expenses as the primary and secondary numeric fields for DIY projects, and selected distance and time for race training. We considered supporting a percentage of progress (e.g., how far along a DIY project is, percentage of planned miles in a race training routine), but we opted for time because it is easier to monitor and is not subject to changes in plans (e.g., a project being less far along than anticipated, more or less training required).
- **Description data**: a text title and description of the chapter. Yarn provides a few suggestions for what might be interesting to write about in these fields.
- **Minor data**: data which may be contextually interesting in a chapter but is only loosely associated with an overall measurement of progress. Drawing from our formative interviews, we included weather information in race training stories and emotion in DIY stories. These data may help explain a moment (e.g., why a run was hard, what is important to learn from a DIY picture), but they might not have a strong influence on how the story progresses.
- **Date**: when the chapter being logged occurred. We include this field to allow people add chapters later, rather than assuming all chapters are written the same day they occurred.

Formative interviewees expressed a range of data preferences, so we made all data fields optional in Yarn. As suggested in prior work, we connect with other apps where people collect data to ease the process of creating chapters [9]. Photos and videos are loaded from the camera roll. Runs are imported via the Strava API (https://strava.github.io/api/) for race training stories, with weather via the Dark Sky API (https://darksky.net/dev/). Runs are plotted on a map via Mapbox (https://mapbox.com/api-documentation/). To avoid redundancy in the story, storytellers cannot create two chapters with the same run or photo. Yarn automatically fills out as much data as possible. Race stories use the Strava and Dark Sky metadata to autocomplete runs, date, distance, time, and weather. In DIY projects, the date field is set from photo or video metadata.

## 4.2  Visual Templates (DS2)

Participant desires to emphasize and explain important moments inspired us to explore different methods for annotating Yarn's

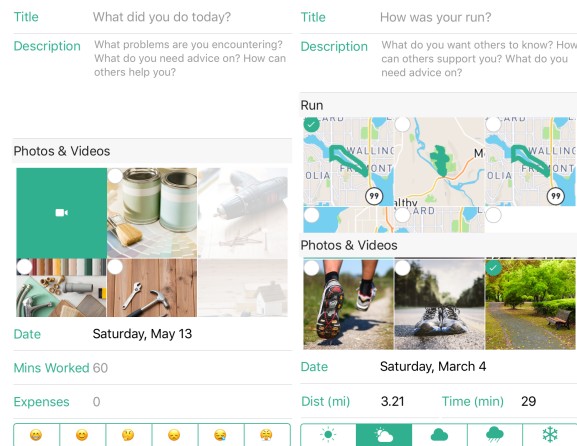

Figure 1. Yarn supports logging five categories of data. To ease the entry process, Yarn infers fields like date and distance based on selected photos or runs and offers writing prompts for description fields. Content shown is illustrative.

visual data with the numeric data. We were particularly inspired by how running apps annotate photos and routes with information about distance and pace. Each annotation, or *visual template*, connects the logged data to a sharing goal. Visual templates also surface how the full story is progressing by presenting the total numeric data (bottom-right of each template, e.g. "*Total hours worked: 5.0*") and how many chapters have been created (top-right, e.g. "*Springfield half marathon: chapter 6*").

We created seven templates inspired by prior work on people's sharing motivations (Figure 2). They emphasize:

- **A question** for when the goal is *information or advice*. This template was added later in our design process.
- **A hard time** making progress, for moments where *emotional support* might be desired. This template extended our initial idea of the personal cost of progress to more specifically request emotional support.
- **I'm back!** Describes how the accomplishment intersects with people's everyday lives by pointing out the time since the last chapter, also designed for moments where storytellers might desire *emotional support*. This template followed early ideas for a template highlighting the challenges overcome.
- **Today's effort**, designed to align with a desire to *share an achievement* by highlighting the progress which was made.
- **My journey**, summarizing all the chapters so far to support a desire to *share an achievement* of how much progress has been made. The template is divided into squares of visual data from each chapter. Stock images relating to the accomplishment make up the remaining squares.
- **A long run** relative to other runs logged, designed to align with a desire to *share an achievement*. We included this template for race training stories only. In the formative interviews, participants wanted a design to reflect when they ran a personal best. We felt the DIY parallel (e.g., a personal minimum or maximum amount of time spent) was not a good measure of progress for most people.
- **Nothing special** when someone did not have anything they specifically wished to highlight. This template was not designed to align with a particular goal. Instead, we designed this template to allow people to create chapters they wanted to record but felt were minor contributions to the overall story of accomplishment. This template drew from discussion around whether every moment necessarily had a sharing goal.

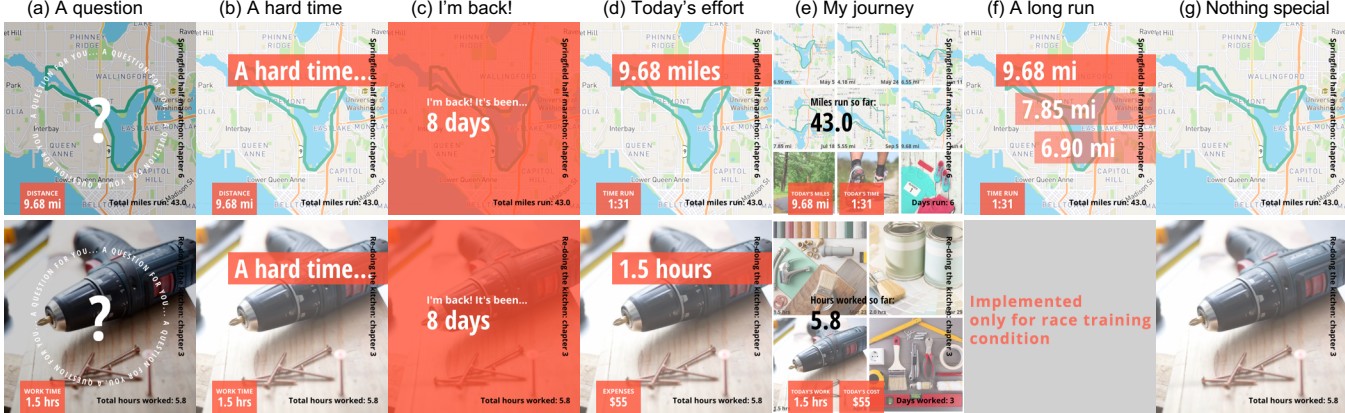

| (a) A question | (b) A hard time | (c) I'm back! | (d) Today's effort | (e) My journey | (f) A long run | (g) Nothing special |

Figure 2. We designed Yarn's templates to support motivations for sharing personal data surfaced in prior work. Template (a) targeted requests for *information or advice*. Templates (b) and (c) aimed to solicit *emotional support*. Templates (d), (e), and (f) were designed to support *sharing an achievement*. (g) supported minor, typically-unshared moments. Content shown is illustrative.

We designed Yarn to support or promote certain templates based on the data entered in a chapter. For example, the **long run** template was only visible when the distance logged was one of the three longest runs. **My journey** was only included as a template option after an author had created two posts. In other cases, Yarn promoted certain templates by defaulting to them. For example, Yarn defaulted to the **I'm back!** template if it had been more than three days since the previous chapter was logged. **A question**, **a hard time**, and **nothing special** were always template options, as was **today's effort** when numeric data was included in the chapter.

Participants in the first field study found the visual templates too rigid. For example, F9run felt templates should be visually distinct: "*they all look very similar…there were 4-5 templates with kind of the same color. It would have been good to have more choice with more diversity*" (F6run and F7run agreed). We therefore modified Yarn to include a set of five template color schemes which complimented one another (Figure 3a). We also added variations in each template's overlay. We added two additional phrasings in the **question**, **hard time**, and **I'm back!** templates, which each use text to direct the audience to the detailed caption. In templates that emphasize data (e.g., **today's effort**, **my journey**, **a long run**), we added options for the template to highlight secondary data fields and minor data fields (e.g., **today's effort** can be used to prominently display expenses and emotion in DIY projects, duration or weather in training for a race). To enable people to tailor a template to their specific experiences, we added the ability for people to write a custom phrase for all templates (via the [Edit] button in Figure 3a).

### 4.3 Description Suggestions (DS3)

To further support people in explaining the importance of different moments, prompts for chapter descriptions contained suggestions for what to write (Figure 1). We designed these hints to prompt the storyteller to consider what they might want to share about that moment. For example, we included suggestions on asking for *information or advice* (e.g., "What do you need advice on?"), *emotional support* (e.g., "How can others support you?"), and *achievements* (e.g., "What are you proud of today?"). Yarn randomly picks three suggestions out of thirteen which highlight different sharing motivations. Suggestion phrasing varied slightly between domains.

### 4.4 Inferring Importance of Moments (DS4)

We presented each story as a chronological feed of chapters (Figure 4) to allow audiences to interpret moments in the broader context of the shared story. To emphasize important moments, Yarn uses heuristics to infer how important a chapter might be, sizing chapters as small, medium, or large in Yarn's feed based on those heuristics. Authors could re-size chapters as desired. We considered enabling Yarn to update a chapter's importance based on whether the author shared it online or by comparing the numerical data to chapters added later, but we decided people might find it unusual if chapters were resized when they revisited the app

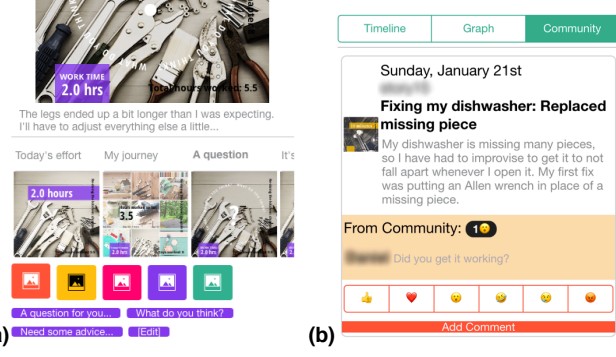

(a)

(b)

Figure 3. We iterated on the design of Yarn to offer more customizable templates and include a community of people working toward a similar accomplishment. Content shown is illustrative.

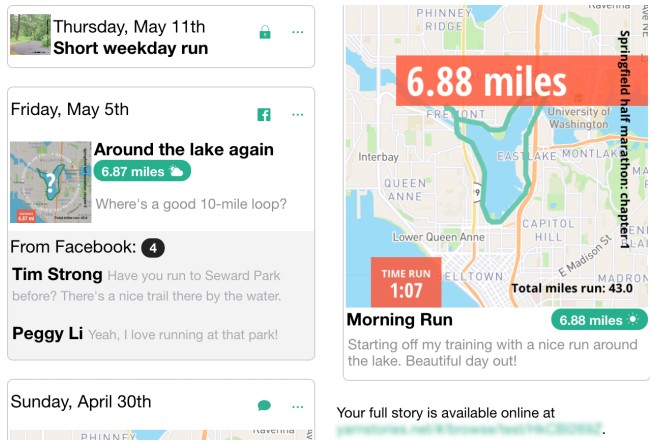

Figure 4. Yarn sizes chapters in a feed based on the numeric and visual data, template selected, and description length. Yarn supports sharing chapters to popular social channels, replicating Facebook comments and reactions in the feed. Content shown is illustrative.

days or weeks later. The heuristics rely only on the data logged in the chapter and the chapters which came before it chronologically.

In decreasing order of weight given, Yarn uses the following to evaluate importance of a chapter:

- The amount of primary numeric data relative to other chapters (e.g., the number of hours worked or miles run).
- The amount of visual data relative to other chapters (e.g., the number of photos or videos added).
- Whether the template was selected before, with no weight for the "nothing special" template.
- The word count of the description data relative to other chapters.

## 4.5 As-Desired Sharing (DS5)

To support people in sharing throughout the story's process, Yarn supports sharing individual chapters as well as the full story. A static URL shows an author's story on a publicly accessible server, and the author can mark any chapter as private if they do not want it to appear in this feed. Many personal tracking apps automatically share moments as they are logged, though people often find this results in uninteresting content [16]. In Yarn, the author can instead share the data associated with a single moment by clicking the icon of a sharing platform (e.g., Twitter, SMS). Yarn then opens a dialog in the platform, including the templated visual data, description data, and a link to the full story. The author can edit the text as they wish for the platform. To serve as a memory aid for social discussion, Yarn records when a chapter is shared to a social platform. It also presents reactions and comments a Facebook post receives (Figure 3) back in the app.

Formative interview participants had varied preferences for platforms on which they expressed interest in sharing their stories. We included the four platforms mentioned most by participants: Facebook, Twitter, Instagram, and SMS. Interview participants were also interested in sharing chapters with Snapchat, but no official Snapchat API existed at the time that Yarn was developed.

Participants in the first study said they would have liked to have a community with other people using Yarn to tell similar stories, echoing a suggestion from prior work [16,46]. For example, F8$^{diy}$ felt "*I would have liked it to feel more like an Instagram style of app where I can see other people's projects and get also inspired and motivated by them and their projects*" (5 other participants expressed a similar sentiment). We therefore added a community feature within Yarn as an additional approach for supporting sharing. The community feature displays the most recent chapters created by other study participants and offers commenting and reaction mechanisms similar to other social networking sites (Figure 3b). Community reactions and comments are replicated in the main timeline view in Yarn, similar to how Yarn replicates Facebook reactions and comments into Yarn's main feed. In addition to the feed of recent chapters from the community, people can look at the complete stories of other community members via a separate page. To protect anonymity, participants select a pseudonym to be used in the chapters and any comments they write. Community engagement is only visible to members of Yarn. Reactions and comments from the community are not shared when chapters are posted on social media nor are they visible on the story's public link.

## 5 EVALUATING YARN'S DESIGN STRATEGIES

To answer our third research question, *can a design help people create content which aligns with goals and leads to support from others?*, we ran two field deployments of Yarn for four weeks each. The first deployment examined whether Yarn's process for authoring a story from data. The second deployment then examined social engagement and responses around content created with Yarn.

To be eligible for either study, participants needed to be actively training for a running race or working on a DIY project. There was no overlap between participants in our formative interviews or either of our field studies. We quote participants with F1-21, again with a superscript for story type (e.g., F1$^{diy}$, F3$^{run}$). The first 10 participants were in the first deployment (e.g., F1-F10), the remainder in the second deployment (e.g., F11-F21).

For the first deployment, we recruited a convenience sample of 10 participants via mailing lists and fliers in a mid-sized technology company. Four participants were working on DIY projects, six were training for running races. One participant (F8$^{diy}$) was working on a DIY project but began using Yarn to record her runs as well (though she was not training for a race). An eleventh participant (male, DIY project) dropped out of the study prior to creating any chapters. We do not report further on this participant. Seven participants identified as female, three as male. Age ranged between 22 and 43 (average 32).

For the second field deployment, we recruited 11 participants through local running and making community groups and email lists. Five participants were working on DIY projects, six were training for running races. A twelfth participant (female, race training) created one chapter before dropping out of the study. We do not report further on this participant. Eight participants identified as female, three as male. Age ranged between 28 and 64 (average 39). Two participants (F16$^{diy}$ and F19$^{diy}$) knew one another prior to the study. Because recruitment drew heavily from community groups, all participants were from the same metropolitan area. Participant occupations ranged from artists to homemakers to analysts to retired.

The two field studies followed similar protocols. We scheduled an introductory meeting with participants where we helped them install Yarn, described the remainder of the study, and conducted a short interview about the accomplishment they were pursuing. Participants completed a three-question survey each week on any bugs they encountered and what they liked or disliked about the app. A longer survey at the end of the study was tailored to the research question in each study. The survey in the first field study asked participants how they felt about components in the design of Yarn through open-ended questions about each feature (e.g., templates, description suggestions, sizing). The survey in the second field study asked participants how they felt about using Yarn to engage with their connections. We also interviewed each participant about answers to their final survey. Interviews were 32 minutes on average (median 31, min 15, max 47). Participants were given a $70 gift card to Amazon. Participants could use Yarn after the study to continue telling their story, but we did not compensate them for doing so.

We told participants Yarn was designed to help tell DIY and race training stories. We did not offer recommendations for when to add chapters, encouraging them to explore the app and use it to write and tell their story as they saw fit.

To understand social engagement and responses around the techniques used in Yarn, we wanted to ensure that participants in the second field study had the experience of sharing their content with Yarn. During recruitment, we therefore asked participants to identify a few friends or family members (at least 1, up to 3) with whom they were interested in sharing their story. After the participant completed the final interview, we sent each social connection a short survey about their experience engaging with the content their participant generated in Yarn, providing them a $10 gift card to Amazon. In total, we contacted sixteen social connections (min 0, max 3 per participant), of which eight completed the survey. We quote social connections with S##a-c, including the corresponding participant number and using letters to

| ID | Planned Achievement(s) | How Far Through Stor(ies) | Social connection(s) | Stories/ Chapters Written |
|---|---|---|---|---|
| F1diy (F, 27) | Home remodelling | Near end, Beginning | | 4/11 |
| F2diy (F, 30) | Making a quilt | Midway | | 1/7 |
| F3run (F, 28) | Half-marathon | Beginning | | 1/6 |
| F4diy (M, 43) | Making kinetic sculpture | Midway | | 2/4 |
| F5run (F, 40) | 10K | Beginning | | 1/10 |
| F6run (M, 22) | 5K | Midway | | 1/6 |
| F7run (F, 24) | 10K | Beginning | | 1/5 |
| F8diy (F, 35) | Creating an inspiration wall Note: F8 also tracked runs. | Beginning | | 2/4 |
| F9run (M, 37) | Half-marathon | Midway | | 2/5 |
| F10run (F, 30) | Half-marathon | Midway | | 1/10 |
| F11run (F, 37) | Full-marathon | Beginning | S11arun (Sister, F, 39) S11brun (Husband, M, 37) | 1/6 |
| F12run (F, 37) | Half-marathon, 5k with son | Midway | S12run (Mother, F, 67) * Spouse * Training partner | 2/28 |
| F13run (M, 43) | Half-ironman | Midway | * Wife * Training partner | 4/13 (split by week) |
| F14diy (F, 31) | Home remodel | Midway | None | 4/5 |
| F15run (F, 30) | Half-marathon | Beginning | None | 1/9 |
| F16diy (F, 28) | Sewing & scrapbooking | End & Beginning | S16adiy (Husband, M, 31) S16bdiy (Father, M, 57) | 4/21 |
| F17run (M, 31) | Half-marathon, Skiing | Midway | * Training partner | 2/13 |
| F18run (F, 45) | 10K | Beginning | S18run (Training partner, F, 51) | 1/9 |
| F19diy (F, 32) | Winter cowl, bullet journals | Midway | S19diy (Husband, M, 35) * Friend | 6/8 |
| F20diy (M, 64) | Jeopardy-style buzzer system | Beginning | S20diy (Partner, F, 52) | 1/5 |
| F21diy (F, 51) | Cabinet for litter box | Beginning | * Housemate * Friend | 2/4 |

Table 1. 21 total participants used Yarn for 4 weeks, 10 in the first field study and 11 in the second. The (*) indicates social connections in the second study with whom the participant stated they shared their story, but who did not respond to our survey.

differentiate between multiple social connections for a single participant (e.g., S16adiy, S11brun).

We again analyzed both field studies through qualitative methods [41]. Interviews, survey data, and the stories participants created were analyzed using a bottom-up thematic analysis through open coding [57], discussing and refining these themes with other members of the research team.

We present the findings of our two field studies by summarizing how both authors and audience members responded to each of the five design strategies in the subsequent sections (6.1-6.5). Table 1 contains demographic and relationship information about the participants in both studies and the social connections in the second study. Across the two field studies, participants created 44 stories and 190 chapters. On average, race training participants created slightly more chapters than DIY participants (9.8 versus 8.1).

## 5.1 Participants Emphasized Visual Data

Field study participants included a range of data types in the chapters they created (Table 2), leveraging both automatic and optional data entry (DS1). Most chapters included some form of

|  | Race Training |  | DIY |  |
|---|---|---|---|---|
| Visual data (%) | Any | 82% | Any | 77% |
|  | Map | 74% |  |  |
|  | One photo | 61% | One photo | 68% |
|  | More than one photo | 21% | More than one photo | 8% |
| Numeric data (%) | Any | 94% | Any | 64% |
|  | Distance | 91% | Minutes worked | 59% |
|  | Duration | 89% | Expenses | 19% |
| Description data (%) | Any | 98% | Any | 100% |
|  | Title | 97% | Title | 100% |
|  | Description | 79% | Description | 90% |
| Minor data (%) | Any | 85% | Any (Emotion) | 70% |
|  | Weather | 85% |  |  |
|  | Temperature | 74% |  |  |
| Date (%) |  | 100% |  | 100% |
| Total chapters created (#) |  | 117 |  | 73 |

Table 2. Participants usually included visual and description data in their chapters, often adding numeric and minor data. More race training chapters included numeric data than DIY chapters.

visual data (82% of running, 77% of DIY). Because participants often loaded their running data from Strava records, numeric data and minor data were very common in running chapters (94% and 85%). These categories were less prevalent in DIY chapters (64% and 70%). However, DIY chapters usually explained their contributions, writing descriptions more frequently than running chapters (90% versus 79%).

As recommended by prior work [16], participants regularly included data beyond the numeric data typical of sharing features. F18run found that visual data valuable to include, saying, "*I just think it works so much better if you have the photos… I just think that the photos make it, otherwise it's just data*". F14diy always included visual data, but, for each chapter, she evaluated whether other data types were relevant to the chapter. She said, "*I took photos of like the steps, like just like one photo per outlet… sometimes if I have it, hours or money spent or whatever*".

The emphasis on visual data, and the anticipated presentation of that data, occasionally encouraged participants to change the activities they completed. F10run mentioned that Yarn "*kind of motivated me to do different trails, since I'm taking photos and stuff it made me want to venture out to different are as.*" In total, 8 of the 10 chapters she logged in Yarn included pictures. This motivation to create interesting visual content by trying new routes continued in her training after the study, and she kept using Yarn for 6 additional weeks.

Other participants appreciated how the range of available data types could help illustrate their progress collaboratively. F2diy tried to highlight how her quilt evolved over the weeks through the visual data, but she struggled because "*once you get to a certain point it doesn't really [visually] change.*" She appreciated how the numeric data demonstrated that she had made progress.

Two social connections indicated that visual data was most engaging. S16adiy felt his social connection's visuals "*were illustrative of the product she was working on and would show applicable aspects of it.*" S11brun agreed, stating that the inclusion of photos in race training "*was a nice added feature that helped visualize the activity*" beyond the map of the route ran. S16bdiy felt the numeric data "*added impact, meaning, and in case I would do it too, valuable information.*"

Participants found the visual data in the community feature helpful for advice and inspiration. DIY authors occasionally felt it was inspiring to see each other's visual and numeric data. F19diy stated "*I feel like not posting a picture, it's like a complete cop-out… like I don't want to see a picture of a paint bucket [the default visual data in the DIY version of Yarn] 'cause that's not inspiring at all*" (1 other participant expressed a similar sentiment).

| | Race Training | | DIY | | Total | |
|---|---|---|---|---|---|---|
| A question | 2 | 2% | 1 | 1% | 3 | 2% |
| A hard time | 6 | 5% | 1 | 1% | 7 | 4% |
| I'm back! | 7 | 6% | 5 | 7% | 12 | 6% |
| Today's effort | 71 | 61% | 16 | 22% | 87 | 46% |
| My journey | 13 | 11% | 5 | 7% | 18 | 9% |
| A long run | 7 | 6% | | | 7 | 4% |
| Nothing special | 11 | 9% | 45 | 62% | 56 | 29% |
| Total chapters created (#) | 117 | | 73 | | 190 | |

Table 3. Participants predominantly selected the "Today's effort" and "Nothing special" templates.

Participants training for races often learned about other places they might run. For example, F18run said, "*I'm always looking for new routes. A lot of [other people's] are near or around [where I run] but not exactly the same. So I was like, 'oh, I could go this way.' Yeah, just a bit of change to make it different*" (2 other participants expressed a similar sentiment).

Although visual data primarily drove interest, social connections felt other data types added to their awareness. S16bdiy felt F16diy used the emotion data field to "*convey her enthusiasm along the way.*" S12run described how weather data added to what F12run shared with her: "*[she] usually shared how she felt when her alarm was going off at 4:45a.m. and what the weather was doing. I loved the times she was running while it tried to snow.*"

## 5.2 Visual Templates Constrained Creativity

On average, participants selected 3 of the included visual templates (DS2) during the study (min 1, max 5). Participants overwhelmingly picked the "today's effort" and "nothing special" templates (46% and 29% of templates selected; Table 3). Some participants, like F7run, felt these two templates depicted everything they wanted to collect and share: "*I pretty much only stuck to the standard [today's effort] template, it just had everything I needed… [adding chapters] wasn't really a creative exercise for me.*" For other participants, the choice was motivated by aesthetics. F4diy said, "*the templates are killing me. I really want one that doesn't touch my image content at all.*" F1diy agreed, "*often when I need to ask a question, there is also an image that communicates [my question] …I wouldn't want to put so much text on top of the image because I want people to study the image to tell me an answer.*" DIY participants seemed more concerned about aesthetics. They wanted to emphasize the quality of their project and allow pictures to be compared to measure progress. Other templates, such as the "I'm back!" and "a long run" template, were used with some frequency, but likely only made sense for moments where those points were in focus.

Participants in the second field study made use of the template customization features we added in Yarn's design iteration (Figure 5). Seven of the eleven participants created at least one chapter with a template color other than the default (max all 4 non-default colors, average 2.3 additional colors used per participant). Eight participants created at least one chapter with an alternate phrasing or custom phrase (max 3 chapters with an alternate or custom phrase, average 1.08 alternates per participant). F11run edited the **I'm back!** template to highlight a return to running after illness (Figure 5a) while F16diy manipulated the **today's progress** to display a custom phrase (Figure 5b). All participants except for one (F17run) created at least one chapter which used the added customization features.

Participants appreciated the additional flexibility provided in the new templates. F15run appreciated the color and template choices, saying, "*I liked the colors, so I thought that was cool. How there's different formats on there that you could kind of highlight different things, like you could highlight how much you ran in the week…or*

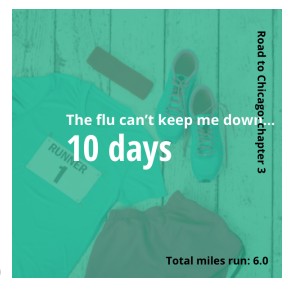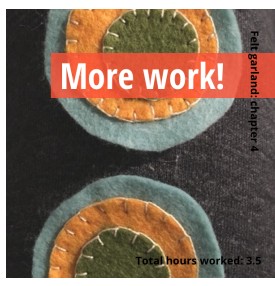

(a)  (b)

Figure 5. Participants in the second field study appreciated the customization options introduced in the design iteration, but still felt the templates limited their creativity. (a) is from participant F11run, (b) is from participant F16diy.

*what the weather was like.*" F18run appreciated the different data options, saying "*I liked the different choices…distance, pace, what are the sunny days or rainy days.*" F12run discovered the ability to edit a template's text midway through the study. She "*made a comment in some survey that it'd be nice for the different templates to be able to enter things, but then I noticed the next week that you can… I liked that.*" Three social connections mentioned they "*liked the variety of templates*" (S18run), suggesting the variations "*kept things interesting*" (S12run).

Although the participants valued the increased flexibility, the customizable templates still did not provide options for editing the visual representation of their chapters. F11run appreciated the different choices, but wondered whether they would scale to longer stories, "*I'm thinking about when I had to do like these daily posts for 90 days… having apps where I can change the colors and the fonts and the shapes kept it new and fresh for me, so I was wanting to use that a lot.*" Participants generally did not feel that the templates achieved the design goal of helping them explain the importance of moments to their story. Race training participants imagined a template could highlight more specific accomplishments, such as "*if it was a PR [personal record]*" (F5run) or what kind of run it was "*if I were trying to do a distance run, say 'distance run', or like a 'short run', or some of them are like, 'hill training'*" (F3run). F6run felt "*if a run was hard, I might want [the template] to say something more specific about it.*"

Participant suggestions indicate a preference for more still more flexible annotation tools, such the ability to arbitrarily place and size text in Snapchat and the story feature in Instagram, rather than Yarn's template-driven model inspired by filter options in Instagram. For example, F11run suggested "*There were several times where I took a picture on a run…and it turned into a Ninja turtle with the thing [colored bar from the template] over my head …In other apps, I've been able to just grab that bar and like drag it down to the bottom so it works.*" F1diy described a similar idea, saying "*I tend to be more on the minimalist side, so I might just have the text displayed more beautifully to say 'my bathtub arrived!'. I have some friends who are really into emojis and might cover half the picture with emojis and smiley faces.*"

## 5.3 Description Suggestions Were Often Ignored

Participants wrote descriptions of a sentence or less for most chapters (55% of running chapters, 61% of DIY chapters). These descriptions rarely aligned with the template prompts we provided (DS3). When they did, they tended to reflect on their feelings about the chapter's content. For example, 19 running chapter descriptions explained how the participant felt before, during, or after the run, such as "*10 miler was tough but got it done*" (F13run). Similarly, DIY participants occasionally explained how they felt about their progress, "*I'm so proud!! i love these little wobbly toys.*" (F21diy,

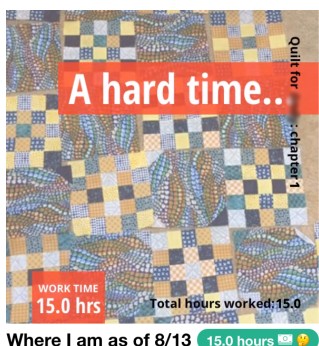
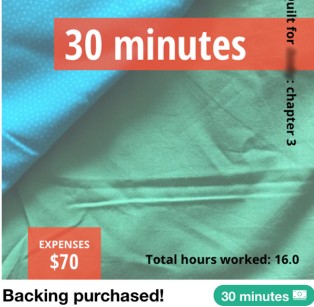
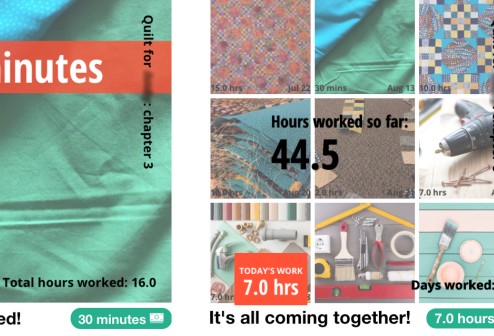
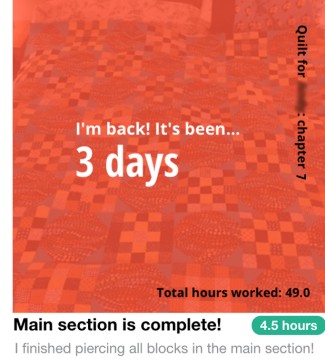

**Where I am as of 8/13** `15.0 hours` 😊📷

As I start my story, I thought I would give a summary of where I am in the process. I began planning a quilt for my husband on July...

**Backing purchased!** `30 minutes` 📷

Since I don't have time to quilt during the week, I have to make due with online shopping. ▪▪▪ liked my idea of a Kelly green backing for the ...

**It's all coming together!** `7.0 hours` 😊

Finished the checker border and 4 strips of the 9 main body rows! My mom helped me to decide to add strip of dark fabric between the

**Main section is complete!** `4.5 hours`

I finished piercing all blocks in the main section! Now I have to cut the divider and attach it and the border to the main section. Getting closer!

Figure 6. The description prompts encouraged F2diy to explain her process of making a quilt and the challenges she faced as she made progress.

3 others) or what they were finding challenging, "*There is more to buzzing in to Jeopardy than I realized… Next need to decide to implement a complete or subset of the buzz in rules as V1.0.*" (F20diy, 5 others). A few participants expressed hopefulness about their progress (e.g., "*I finished piercing all blocks in the main section! Now I have to cut the divider and attach it and the border to the main section. Getting closer!*" F2diy) or concern about the next stages (e.g., "*need to squeeze in some more runs next week!*" F7run). When participants provided context suggested by the prompts, social connections appreciated these descriptions. For example, S11arun liked how the descriptions in Yarn helped her "*get into her head here, how she felt, victories and frustrations – this is important.*"

Instead of following the prompts, participants primarily used the description field to add extra context about the moment they were sharing. Running participants tended to describe three factors: the route, such as "*ran a loop around Manitou beach*" (F5run, 13 other chapters), when they ran, such as, "*quick run after a game of basketball*" (F10run, 7 others), or who they were running with, such as, "*the third Tuesday of the month means Territory Run Company's Sunrise Run around Discovery Park*" (16 others). DIY descriptions overwhelmingly explained the progress which was made, such as "*Starting the project by measuring out the space to buy tiles*" (F1diy) and "*Ironed fabric. Squared it up. Cut all my squares.*" (F16diy, 39 others).

When asked, a majority of participants said they tended not to notice the description prompts or intentionally chose not to follow them. They therefore had little effect on what participants wrote or their thoughts. F18run described, "*sometimes I think I remember reading a couple of questions but not always.*" Others, such as F19diy wanted to explain their progress rather than try to explain the moment's importance. She said, "*[the prompts] didn't quite match what I was trying to do… Like 'what did I do today?' Versus like, 'what are you working on' or 'what did you just make?'… I don't know, it's just like, it's too, it's too big of a question.*"

A few participants felt that Yarn's description prompts motivated them to reflect on a moment's importance and add that detail. For example, F7run felt the prompts encouraged her to think more about how her run went: "*some days, [the prompts] did help me reflect on my run, which was nice.*" F2diy agreed, adding "*the prompts were good… having those fields where you could put what you were working on and what things you were actually encountering… it just focused me and allowed me to write a lot.*" Figure 6 shows four chapters from F2diy's story. Other participants ignored the prompts altogether, writing "*based on my feeling*" (F9run). But even those who ignored the prompts still felt they encouraged authoring interesting content. F1diy felt "*I thought [the prompts] were actually*

good ideas for what I might write. I didn't always follow them, but I did usually read them.*"

## 5.4 Automatic Emphasis Went Mostly Unnoticed

The 10 participants in the first field study were asked their opinions on Yarn's automatic emphasis (DS4). Of those 10, half (5) indicated that they had not noticed the range of sizes at all. The algorithm did not create substantial differentiation in how some people's moments were sized, such as F10run "*[I'm] not sure if it really ranked my chapters. All of them appear open [large], but one.*" (2 others expressed the same sentiment). Of the participants who did notice, some felt the automatic inference was unusual: "*I noticed that they were different sizes, and I always thought it was a little odd, but I never actually stopped to think about what the different sizes were for*" (F1diy, 2 others agreed). F1diy would have preferred to either directly indicate how important was, or "*to be honest, I don't know to what extent I want the sizing… the size doesn't draw visual interest.*"

However, the few participants who did notice appreciated how Yarn inferred importance from the data they entered. F5run liked how the sizing emphasized her longer runs, "*there's so many standard runs you've got to knock out three times a week, and then once a week you have a more challenging, long run… so it is nice to highlight that*" (3 others agreed). F2diy agreed, saying "*it seemed like the entries where I actually spent time on the work, they were the ones which were larger*" (1 other agreed). Though DIY participants thought time worked was a reasonable measure of importance, they also had other measures in mind (e.g., how they felt, how far along they were).

Participants overwhelmingly chose to share individual chapters rather than the link to the feed, so only a few audience members saw Yarn's automatic emphasis. None expressed noticing differences in how chapters were sized. Participants in the second field study, who had the community feature to view each other's stories, indicated that they paid more attention to aspects other than size to determine what chapters they wanted to read. F19diy said, "*probably just the titles that really got me interested… it was just like, 'oh, I wonder how that turned out'.*" Others, like F13run, paid more attention to chapter's content than the size. She said, "*I saw people that had a map, where they ran with stats, but also a picture that they took on the waterfront or something.*"

## 5.5 People Received Encouragement, but Desired More Social Engagement

The majority of engagement around Yarn-generated content was encouragement, both among social connections and in the community group. F11run described, "*both my sister and husband*

*thought it was interesting… their responses were supportive*" (6 others expressed similar sentiment). F15[run] found that other participants used the title, description, and templates to garner more encouragement, "*some people put funny titles or commented on what was hard for them, and that makes you want to encourage them more if they maybe show some personality or some progress or something in their little comments*." By sharing as a story, Yarn also occasionally made the audience aware of otherwise invisible work the author had been doing. F13[run] described that his wife found out, "*what she didn't realize was that I do a lot of running around work, she never really sees or hears about those. So she found that to be interesting*" (F13[run]).

Though all participants identified friends or family members with whom they wanted to share, two participants (F14[diy] and F15[run]) did not share their story with their social connections. They both felt that content created in Yarn had too much overlap with content that they already shared with their social connections via other platforms (e.g., running data on Strava for F15[run], progress photos on Instagram for F14[diy]). F11[run] posted photos from her runs to Facebook during the study, but only shared Yarn content with her recruited social connections. F11[run] and F14[diy] were both turned off by the restrictions imposed by Yarn's templates. F15[run] felt Strava was sufficient for how she wanted to tell her story, saying "*I didn't really share it a second time since it was already on Strava.*"

Of the participants who did share, most shared their chapters with their social connections in-person soon after creating the chapter (DS5). Two participants (F16[diy] and F20[diy]) shared nearly all of their chapters to one of their social connections via SMS. Six other participants used the SMS feature at least once. Only one participant in the second study used Yarn to share a chapter to social media (F18[run] shared to Instagram once during the study).

Although authors hesitated to share their progress to broader social networks, social connections particularly appreciated being able to see the chapters of the author's progress. S11a[run] felt that "*I got more details than I normally would about the day-to-day training*". In particular, S11a[run] learned that F11[run]'s training had not progressed smoothly: "*I hadn't realized her recent setback in training and it was interesting to see how she described it on Yarn, and then be able to talk to her about it.*" Social connections added that sharing chapters through Yarn gave organization to the story. S16a[diy] appreciated that Yarn provided "*regular updates on particular benchmarks with beautiful and concise information. It bundled the information very well instead of hearing things in a haphazard way.*" F19[diy], who knew F16[diy] prior to the study agreed, adding "*even things she hadn't sent me in a text message I saw on [Yarn], and I'm like, 'oh, you completed it.' So it was fun to see the progress she made.*"

Despite appreciating being able to follow progress, some social connections felt the stories in Yarn did not convey the author's motivation or the scale of the accomplishment. S16b[diy] said, "*it was nice to see [F16[diy]]'s progress, but I wasn't sure of the context – why was she pursuing a particular project, what the scope of it was, etc.*" F11[run] felt Yarn could have better supported her in demonstrating how individual training moments contributed to her accomplishment. She said, "*[my social connections] might've been more engaged if there were reasons to keep interacting, like watching for milestones or PRs on a progress bar or something.*" S11b[run] agreed, stating "*progress or milestone markers would have been nice.*" People's stories often deviate from their plans, so it remains challenging to support presenting this progress. But participant feedback suggests that sharing achievement of critical milestones could add more value than cumulative distance or time.

The 11 participants in the second field study had access to the community feature. Of these 11 participants, 5 wrote at least one comment or sent a reaction in the community feature (responding to a max of 4 chapters). In total, 10 chapters written by 4 unique participants received reactions or comments. 3 participants both sent and received reactions or comments.

In addition to learning from what others were doing, participants felt the community added a sense of accountability to continue making progress toward their accomplishment. F15[run] felt that "*seeing other people running adds like a little motivation for me to keep up with running.*" F16[diy] agreed, stating "*I got one or two hearts, and I think it's fun to know that someone is looking at your stuff and a nice boost to keep going.*" Though this is similar to prior work involving shared challenges [18] and activity goals [8,60], we note that in our study participants pursued a wider range of accomplishments (e.g., different races, more varied DIY projects).

Because their accomplishments were dissimilar and participants did not know one another, many participants were reluctant to engage or had a hard time deciding how to respond. F12[run] felt, "*it was nice to see other stories out there, but I didn't really dig in too deep. I definitely went through… someone's training for [the] Chicago [marathon], that's cool. I hear that's a good marathon or whatever… it just feels weird to me to go on and be like, 'oh, someone I don't know is training for a race I'm not running, I should go see what they're doing.' Maybe I shouldn't care so much, we fall in the same boat [of runners].*" F20[diy] felt similarly, struggling to see how he might learn from, provide advice to, or be inspired by some of the other DIY projects. He said, "*some people were building a cat house, other people were knitting, other people were doing some quilt thing… I didn't really care about those much, but if someone had been doing a home security thing with cameras or electronics, I might have been like, 'oh, that's interesting and something I might want to do myself.'*" Three participants suggested that Yarn follow a more traditional social network setup where people can follow one another. For example, F15[run] said, "*I would prefer if it was just that my friend could also have the app and we can see each other's [runs] on there… in my mind, then the people are looking at what I'm posting because they're interested in seeing it rather than it be more motivated by me sharing it.*"

Overall, most participants wished they had received more response from the community. F14[diy] felt that having a social response would have been "*a little more encouraging to continue on with whatever story or whatever project I was working on and actually follow through with it to completion.*" But participants felt the varied accomplishments and not knowing the others were barriers to responding. F12[run] said, "*I don't want to be the weirdo that's like, 'here's your thumbs up'… but once that gets going and that's kind of the social norm of it, [then] it wouldn't be a big deal.*" F19[diy] felt the reaction options provided a barrier to further conversation. She said, "*the different emojis made it kind of like a hard stop. Like, oh, you do a face and the conversation is over. I think I'd actually prefer if those faces weren't there… that way would've made it more of a conversation, or the comments probably would've had more usage.*"

To overcome these barriers to participation, some participants recommended grouping strangers by more specifically similar accomplishments, such as the same race or DIY projects involving the same materials. F11[run] felt, "*if it was people that also did similar things, so they ran the same route or were training for a race at the same time… or if there were things that might have similar backgrounds, like, 'hey, I'm breaking in a new pair of shoes'… I think that would draw me in.*" 2 other DIY participants agreed, with F21[diy] saying "*maybe if it was almost like a forum for particular crafts, so you knew your audience was just people who are doing the same sorts of things. Which I guess [Yarn] kind of is actually, but even within the app there's a lot of different things.*"

## 5.6 Yarn was Useful for Documenting

Despite our goal of promoting sharing, 14 participants mentioned using Yarn primarily to track progress for their own later reflection and reminiscence (e.g., documentary informatics [13,50]), rather than to get support, advice, or feedback from peers or social ties. F19[diy] felt that Yarn was "*documentation for me…it was good for [my] process, just for me to be mindful of*." Although F12[run] appreciated the feedback and support from her close ties and the community, she described training for her race as "*my own private journey*." F13[run] similarly felt the record he created in Yarn was just for him, saying "*I didn't like or comment on anyone else's stuff because I was pretty much under the impression that we were all just doing our own thing*."

Some participants used Yarn's record to help them monitor how far along they were or how much progress they had made so far. F3[run] said "*the graph/summary page is pretty useful just to get an overview of my training*." F2[diy] agreed, adding "*[Yarn] gave me a sense of progress and an idea of how much resources I've spent in the process*." Others were more interested in reflecting on how they felt in the moments they logged. For example, F15[run] used the thoughts she recorded in the description fields, alongside the numeric data, saying "*being able to reflect on not just like the physical things you're going through, like the time it takes and all that, like having some sort of mental or emotional response. It's kind of cool to be able to see that*." F17[run] used those records to think through how he might improve: "*if I take [what I did] down for the future, hopefully I can improve upon it…how did it feel had I eaten well the night before a day of a run or something?*"

Although participants were primarily interested in documenting for themselves, they suggested that they still wanted to share with others at different stages of the process. For example, participants felt they would be more interested in sharing their story to social media once it reached completion. F8[diy] felt her project was "*not done, so it's not something to be proud of yet*." F1[diy] agreed, stating that "*I might share a before and after photo at the very end of a project, but I don't want to 'show off' by showing people how much work I did or bore people with all the stages*" (5 others expressed a similar sentiment).

## 6 DISCUSSION

The design of Yarn operates as an aggregator of tracked data, translating data collected in another app (e.g., Strava for running) to a story. This structure has the benefit of supporting a range of data collected from other applications to tell stories through other domains, such as incorporating Duolingo data to tell the accomplishment story of learning a language. But as some participants suggested during the study, Yarn may not need to be a standalone app. Rather, structured authoring could be added to personal tracking apps (e.g., Strava, Mint) or to social networking platforms (e.g., Facebook, Instagram, Snapchat).

By re-framing sharing as storytelling, Yarn's design highlights how personal tracking apps can help people receive support while a story is in progress, in addition to celebrating its completion. Current social feeds within tracking apps highlight recent events isolated from goals, such as the latest runs of followers on Strava. They could instead emphasize progress made toward accomplishments, if a person has designated a goal. Designers of social network sites should continue exploring ways to support connecting moments that contribute to a larger accomplishment. One example is the ability to continually add to an album on Facebook, updating the album's followers via their timeline. Drawing inspiration from "story" features (e.g., in Snapchat, Instagram, and Facebook), designs could group stories by the varied accomplishments an individual is pursuing and could persist stories according to the accomplishment's timeline (i.e., in contrast to the ephemerality of current features).

We take away some key points from our deployments of Yarn. First, participants were primarily motivated to visually express their stories rather than a desire to share numeric data they collected about how much time they spent or how long they ran. Second, although participants did use a variety of visual templates, they selected plainer templates for most moments. Third, participants felt that their stories were primarily valuable as records for themselves, but they appreciated engagement when it was received.

## 6.1 Targeting an Interested Audience is Challenging

Audiences appreciated seeing participant's intermediate steps, and participants enjoyed learning from other's progress in the community feature. Although few audience members saw a participant's feed, participants appreciated being able to reflect on their progress. Participants moved beyond numerical summaries and system-generated content, customizing a range of visual templates and explaining a moment with the description fields. Most participants wished Yarn had done more to emphasize the moments they felt were important, or that it had given them the ability to emphasize them visually.

However, participants felt that some of Yarn's design strategies were ineffective or even counterproductive. They also found the visual templates and description suggestions too restrictive, usually selecting the most basic visual templates and often ignoring the description prompts altogether. Although participants did receive some encouragement from peers and social connections, most were still hesitant to share in-progress accomplishments. Most also desired more support and advice than they received.

Previous work highlights that when sharing personal data, both designers and users can find it challenging to identify an interested audience [16,43,44,46] and convey the data's importance [16,37]. When we designed Yarn, we expected that explaining accomplishments to close friends and family members would help people get the support and advice they desired. Although participants appreciated connecting with those audiences, we learned that a storytelling tool is ineffective if the storyteller cannot choose the appropriate audience for the moment. The peer community was able to provide different motivation and advice than closer ties less familiar with the domain could, aligning with prior work differentiating the questions people ask of Q&A sites versus their social networks [42] and the different online spaces in which people choose to share health information and seek support [46]. For many participants, telling the story to others was not as important as preserving a record for themselves which explained what they did and how they felt.

Participant responses to the design strategies in Yarn suggest new approaches for simultaneously supporting storytelling through personal documentation, sharing with peer communities, and sharing with close ties unfamiliar with the domain. Informed by our findings, we discuss tensions to balance when researching and designing future tools.

### 6.1.1 Facilitate Documentary Informatics *and* Social Support

Much like the style of content common in journals, Yarn encouraged participants to log how they were feeling and the importance of the moments they were experiencing. Though in practice, participants often ignored these encouragements. When participants did act on this, the combination of emotional state with visual and numeric data about what a person experienced, Yarn facilitates people in revisiting their histories over the long-term. Yarn's guided authoring approach of combining emotion,

description, and data helps the tool be useful for later personal reminiscence, even if a person tracking has no interest or desire to share their story with others. This parallels how people's memories logged on Facebook can facilitate "backstalking" (i.e., looking at a person's historical social media posts), alone and with others [52].

The desire for social support often motivates people to track and share [16,55,58] and is an effective tool to increase motivation and sustain adherence to tracked goals [11,18]. Although participation in the study likely encouraged participants to sustain use, a few participants described feeling that the social support provided them with additional motivation to continue, and they appreciated the encouragement they did receive. Participant's documentary informatics motivation meant that many felt that Yarn's content was as much "for them" as "for others". Participants therefore perhaps felt ownership over their story's structure and presentation within the scaffolding provided by Yarn. They often used the data fields to embellish their record and selected the plainest templates, rather than use those fields to communicate a need to a potential audience. This suggests that perhaps the approach used to develop Yarn's templates could be replicated with a focus on additional templates supporting documentary-focused tracking.

Designers of tools supporting storytelling with personal data face a tension in supporting both the self and social connections. For the self, reminiscence and self-reflection could be served through a range of data, rich descriptions of the event, and plain visual styling [48]. For social connections, minimizing data and explaining the event's importance are more important [16]. Our study suggests that mechanisms designed to support one group's needs can come at the detriment of the other. Future approaches could include editable defaults, such as editing the position of visual template elements. Recommendations which can easily be ignored, such as Yarn's description prompts, can also help support both groups. People's goals for collecting personal data often change over time [17], and their intended audience could change as well. Re-framing content generated with a different audience in mind poses additional challenges. This challenge might suggest questions of how a tool like Yarn might re-purpose fields in a template when a person's goal for that content later changes (e.g., migrating values as defaults in a new template better suited to the new goal).

### 6.1.2 Balance Structure with Flexibility

Many participants felt Yarn's visual templates were so structured that it stifled their ability to tell their story how they wanted. Though the literature demonstrates that flexible presentation can be mediated through letting authors draw their data on paper [1] or with the assistance of an expert trained to curate data [13], neither approach scales to fit the needs of many people who want to share their data. A design aiming to help people flexibly author moments themselves should aim to ensure content explains what someone did, why they did it, and how they feel, while still providing enough guidance that the content remains visually compelling.

One approach that designers could leverage is the sticker and filter metaphors in current "story" features as opportunities for incorporating data. For example, a Snapchat filter could incorporate someone's running route, or a sticker could allow someone to label a snap of a table they built with how long they spent working on it. These data-driven annotations can help explain the importance of a moment or their motivation for tracking and sharing, concepts which audience members sometimes had trouble identifying. These system-provided stickers and filters add some necessary structure to ensure the content remains compelling, we suspect the range of choices available and the ability to directly manipulate position, size, and orientation would provide authors enough say in how moments are presented.

### 6.1.3 Gauge Audience Interest and Expertise

Prior work suggests that broad social networking sites like Facebook, Twitter, or Instagram might not be an appropriate place for sharing personal tracked data [16,43,44]. Different sharing motivations lend themselves to different audiences [46]. Our findings suggest that close ties and communities with similar accomplishments appreciated seeing intermediate progress, and most people feel comfortable sharing their milestones to these groups. Sub-dividing audiences by more specific goals (e.g., length of race, materials used in DIY project) or typical data collected (e.g., running similar distances on the same days of the week, similar schedules for working on projects) could foster additional interest and opportunity for offering advice. Participants whose accomplishments were still weeks or months away indicated that they might want to share to a broad social networking site when they completed their story, but they felt their progress was not sufficient enough to warrant sharing immediately.

Given that social connections were interested in seeing intermediate progress, story authors may have misjudged how interested a broad social networking audience would be in seeing the steps toward their accomplishments. Alternatively, story authors may have successfully identified the few social connections they had who would be interested in offering support and advice on intermediate accomplishments. Future work on understanding how broader social networks respond to Yarn-like content would help designers determine whether to support or even emphasize sharing accomplishment progress to these networks, or instead emphasize keeping strong ties informed.

Prior work suggests that peer groups similar to the community in Yarn often provide support or encouragement when a person posts [18,43,45,60]. However, Yarn participants often felt they did not have enough expertise about the domain or understand the context well enough to respond to people's chapters. We suspect this occurred because Yarn participants did not have explicit support goals and rarely conveyed informational or support needs in the chapters they created. Perhaps techniques from social translucence [19] could provide encouragement absent an explicit information need, such as surfacing how many people in the audience viewed the chapter, even if they did not explicitly respond.

## 6.2 Limitations

We created and evaluated design principles for authoring in two domains of frequent study, race training and home DIY projects. Though focusing on these two domains can offer some guidance for how designs can support people to tell stories of accomplishment with a variety of tracked data, further study will be important to establishing recommendations in new domains. Our small participant populations enabled us to engage with participant feedback and iterate on Yarn's design. Further examination of how a larger, more diverse population responds to principles for authoring with tracked data will be important to refine these design recommendations.

The second field study characterized the social interest and response to content generated with Yarn by collecting the opinions of close ties (e.g., family members, others also interested in the activity being tracked). Having participants recruit close ties allowed for understanding how audience members felt about engaging with a person's story at multiple points as it unfolded. Weaker ties who might engage less frequently, such as friends and family on social networking sites, will likely have different reactions to generated by Yarn and are worthy of further study.

We did not directly compare Yarn's feed of progress toward an accomplishment to the more ephemeral story features in

commercial social networking applications like Snapchat, Instagram, and Facebook. In most cases, we expect sharers and recipients of stories of accomplishment would prefer some preservation of the story across moments. Participants felt the feed allowed them to better contextualize the importance a moment had in their larger story. However, additional work explicitly comparing an ephemeral strategy to a feed-based strategy would be necessary to understand the tradeoffs of lasting versus ephemeral stories for different types of narratives.

## 7 CONCLUSION

We contribute findings from a human-centered design process examining how to better support people in sharing their experiences with data, their way. Though the Yarn app we developed received mixed reactions from participants, the design and study of it helps inform design goals for future tools. Participants moved beyond the numeric, system-generated summaries pervasive in the social features of today's commercial tracking applications. Instead, participants used Yarn to include visual data and detailed descriptions of the moments they collected. But participants felt Yarn imposed too much structure on how content was shared. Although audience members appreciated seeing intermediate moments, participants remained reluctant to share until after their story was complete. Designers of systems supporting storytelling with personal data must balance a range of tensions. People want to use the content they generate for personal reflection and reminiscence, but audiences desire additional explanation and perhaps less data. Although structure can ease the storytelling process, it can also make the generated stories feel impersonal.

## 8 ACKNOWLEDGMENTS

We thank Sol Choi for help with interaction design, Koko Nakajima for visually designing the templates, Ailie Fraser for piloting Yarn, Jessica Hullman, Karrie Karahalios, and Joy Kim for their feedback on the design and evaluation, and Madisen Arurang, Peter Cutler, Maya Klitsner, and Tre Paolini for managing the second deployment study. This work was funded by in part by Adobe Research, the National Science Foundation under awards IIS-1553167, IIS-1813675, IIS-1850389 and a gift from Facebook.

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
