# OpenReview forum: "Yarn: Adding Meaning to Shared Personal Data through Structured Storytelling"
_graphicsinterface.org/Graphics_Interface/2020/Conference — GI 2020_

### Official Review · AnonReviewer1 · 2020-02-12
**Review of Yarn: Adding Meaning to Shared Personal Data through Structured Storytelling**

**Rating:** 7
**Confidence:** 3

**Review:**

This article presents Yarn a documentation and sharing application for personal activities. Yarn focuses on running and DIY activities. The authors based the design of Yarn on interviews with local people interested in the project. They iteratively developed and deployed an iOS based on their insights and user feedback. And finally conducted a study of Yarn with 21 participants. The amount of work that went into designing and evaluating Yarn is impressive.

# Related Work
The project touches on many sub-fields of HCI, from personal informatics, to research on reflection and reminiscence, to research on social aspects of sharing. The related work does a good job of treating all these topics and drawing relevant insights from the literature.

# Study 1
Section 4 presents an interview study focusing on what type of stories people would want to tell based on personal data.
The study is insightful and the four design guidelines are clearly presented.

# Yarn
The design of Yarn is clearly presented.

# Study 2
Section 6 conducted a field study of Yarn with 21 participants over four weeks. Section 6 would benefit from a more structured presentation of the method as is classically done in HCI papers (Participants, Study Protocol, Data collected, Data Analysis Method). The tables are somewhat overwhelming and hard to parse. Diagrams and graphics could help to better understand the scope of the deployment, and the profile of the participants.


# Remarks and spaces for improvement
- The Introduction raises one overall research question, split into 3 sub-sections, and followed by five contribution (with sub-contributions). This leads to some confusion on the message of the article. I would encourage the authors to structure the questions and contributions to match each other, and to focus on what are the most relevant ones. For instance the type of "data to tell" risks to be heavily influenced by the recruiting strategy, in term of social experiences of participants that may not really be representative of the country the study was conducted in, and much less of other cultures internationally.

- Section 3 is not really needed and could be cut altogether, “human-centred design” is the de-facto approach in HCI, there is no need to present it. The half of section 3 announcing the paper structure could be integrated to the introduction.


# Synthesis
Overall, the article is methodologically sound, and while the results are not ground-breaking they will be of interest to people working on the topic. The richness of the application, and the diverse audiences it caters to might have led to challenges in framing the precise contribution of the paper. Insights relevant to people working on related topics are peppered throughout the article, but a big overarching view of the main contribution is missing.

On the , the authors would probably benefit from getting more familiar with the (Interaction) Design literature, especially on the modes of knowledge production in Design (e.g. designing for the particular instead of the generic [1]) and how to go beyond human-centred-design (see Will Odom’s work for instance). Given the situatedness of in the wild deployment it is quite normal not to find significant effects, or minimal adoption/appropriation. The value of design inquiry lies in reflecting on the detailed idiosyncratic practices participants develop, regardless of generalisability.


[1] Olav W. Bertelsen, Susanne Bødker, Eva Eriksson, Eve Hoggan, and Jo Vermeulen. 2018. Beyond Generalization: Research for the Very Particular. Interactions 26, 1 (Dec. 2018), 34–38. DOI: http://dx.doi.org/10.1145/3289425

---

### Official Review · AnonReviewer3 · 2020-02-12
**Interesting application for social sharing of structured stories**

**Rating:** 8
**Confidence:** 3

**Review:**

This well-written, clear paper presents the design of a mobile application to support the sharing of personal stories using templates that are contextually customized to prompt easy curation and sharing of stories. The design is grounded in a formative study that motivates the requirements.

The resulting design responds to the findings of the formative study to present an app that allows users to quickly choose a template and customize it to create a visual story that can be shared a social website of the user's choice. For example, templates are pre-populated with images and prompt questions based on their theme, and the user can customize the image or add a new one, answer the questions or write something else. Some data is automatically populated (e.g. date) or made available (e.g. map of run). Then the user shares the combined image with friends.

The initial prototype was populated with templates related to running (getting started to run, hard day, etc.) The application was deployed in a pair of deployment studies with real runnings for 4 weeks each. The participants all had an active running lifestyle. Rich field study data about the use of Yarn and the participant feedback including quotes were provided. Some interesting findings, such as visual templates constrain creativity (while providing other functions) were surprising and informative. The description sentences preformatted into the box were also not well subscribed as users preferred flexibility. Finally, there was an unexpected finding that Yarn is best used for personal notes rather than shared. Both the formative study and field study are clearly reported with extensive evidence of participant feedback.

Overall, I liked this paper. The accompanying video was interesting. as were the supplied supplemental materials. The paper is a bit longer than necessary to tell the story, but there were no parts I felt were egregious as the longest sections included quite a few participant quotes which was helpful.

The negative of this paper is the level of contribution in terms of what new, generalizable HCI is coming from the work. The results may not generalize to other story-sharing support systems.

---

### Official Review · AnonReviewer2 · 2020-02-16
**A thorough user-centred investigation, some points important to add (that prevent a higher score) but possible to address.**

**Rating:** 7
**Confidence:** 5

**Review:**

The submission presents a user-centered investigation of how to help people involved in long-term goals (running training, DYI projects), create and share stories of their progress. The goal for creating these stories can be either personal, sharing and receiving feedback, or both. The paper follows a very thorough design methodology, starting with interviews with interested parties (23 participants) and deploying the designed mobile app (Yarn) in a field study (21 participants). Yarn as far as I can tell, goes beyond existing social media apps in that: it allows the creation of persistent stories, story posts are divided in goals (rather than one unified feed), and there is support to use different templates for structuring the story depending on the author’s needs (goal may be to get feedback, request emotional support, to inform about progress or of an achievement). The paper discusses in detail several findings of the use (or not) of the templates and the response from social peers of the authors.

The strongest point of this work is the methodology followed. The choice to involve end-users in the form of interviews and a field study is appropriate, the process is well described, and the analysis well reported and discussed. This is really admirable. I was also very pleasantly surprised by the honesty of the work. The results, including negative ones (like the fact that participants worked with few templates and found them generally restrictive) are objectively presented and discussed.

The creation of the story templates that are “goal driven” is not something that I have seen in such tools before. This choice was well grounded from the interviews. It was thus fairly disappointing  to see that in the end very few of these templates were used in practice. Some of the explanations provided in the paper (templates too rigid, their visuals may prevent people from showing the image, they did not highlight important milestones, etc.) make sense. Given that the most popular templates are “today’s progress” and “nothing to report” I wander how much of this is the actual template and how much it is that the story is still being built and as such harder to reflect upon-  i.e., other templates may make more sense at the end of the story. I would have liked to see a summary (in a table?) of the types of templates used per participant and ideally when they were used in the story (chronologically). I would expect for example that posts like “long run” or “my journey” are naturally less frequent because they happen at the end of the goal /story (or are just rare). I believe the authors should provide this information to contextualise the use of the templates.

I also appreciated the attempt to ask authors specific questions in order to suggest templates and text for their posts in the story. While this seems to be in practice unsuccessful in determining what msg the authors want to pass, I found it very interesting that some participants became more reflective of their stories nevertheless.  I believe it is worth stressing that these questions may be a practice worth adopting, not for detecting/suggesting the text but to help authors distill what is important about their post they are sharing.

It is indeed surprising that despite all this effort, the audience did not react to the posts as much as the authors of the stories wanted them to. The reasons given in the paper (audience not sharing/understanding the goal) is very convincing. But I am wandering if it is also related to the lack of use of the “question” and “emotional” support templates, (or if this is unrelated and they gave up on the use of the template). Again, some information about how often and when the templates were used would have helped.

I would also have liked for the paper to stress these more surprising findings (discussed above). Now they are lost in the numerous observations reported (I may have even missed some - these are the ones that resonated to me). A summary, in the end of the observations, in the discussion or in the conclusions, could help readers with some key take-aways.

I would have also liked for the paper to clearly state the technical/feature contribution over existing work and systems. In my summary (1st paragraph) I have listed the ones I felt are novel, but I may be wrong,  it would be good if the paper highlights these.

Finally, the paper should acknowledge other forms of expressing personal stories in the related work. For example there is extensive work in data visualization for personal data and reflection through personalized visuals [a][b][c], work on visual templates for stories [d]) (that comes from a long line of work in data storytelling, a recent book on the topic [e]). While this submission deals with stories that mix numerical and pictorial information, and focuses on construction, I believe this work is relevant and worth mentioning.



Overall, my recommendation would be to accept this work with the following additions:
- A table or visual showing when and how often the different templates were used in a story (possibly per participant).
- Highlight the most surprising findings (for me it was the list of all the items/findings I present in the paragraphs above, but there may be others I missed). I believe these are worth iterating in condensed/summarised form somewhere (in the conclusions or end of the discussion).
- Stress what is novel (in terms of design/features) in Yarn. I have provided my interpretation in the first paragraph of the review but may have missed something or misunderstood something as being novel.
- Add in related work relevant papers from data storytelling. I provide some that I believe are very relevant but encourage the authors to read more on the topic.


Minor
====
- p7-8 I found the need for the templates “a hard time” and “I am back’ a bit hard to understand from the interviews, it may be worth expanding a bit more
- p1 allow people to share _with_ others who
- p12 and p18 at the bottom of the page is the title of a section that should be in the next page
- p15 connec tions => connections


[a] Nam Wook Kim, Hyejin Im, Nathalie Henry Riche, Alicia Wang, Krzysztof Gajos, and Hanspeter Pfister. 2019. DataSelfie: Empowering People to Design Personalized Visuals to Represent Their Data. In Proceedings of the 2019 CHI Conference on Human Factors in Computing Systems (CHI ’19). Association for Computing Machinery, New York, NY, USA, Paper 79, 1–12. DOI:https://doi.org/10.1145/3290605.3300309

[b] Alice Thudt, Uta Hinrichs, Samuel Huron, and Sheelagh Carpendale. 2018. Self-Reflection and Personal Physicalization Construction. In Proceedings of the 2018 CHI Conference on Human Factors in Computing Systems (CHI ’18). Association for Computing Machinery, New York, NY, USA, Paper 154, 1–13. DOI:https://doi.org/10.1145/3173574.3173728

[c] http://giorgialupi.com/dear-data

[d] E. Segel and J. Heer, "Narrative Visualization: Telling Stories with Data," in IEEE Transactions on Visualization and Computer Graphics, vol. 16, no. 6, pp. 1139-1148, Nov.-Dec. 2010.

[e] Data-Driven Storytelling (AK Peters Visualization Series) 1st Edition by Nathalie Henry Riche (Editor), Christophe Hurter (Editor), Nicholas Diakopoulos (Editor), Sheelagh Carpendale (Editor)

---

### Meta-Review · Area_Chair1 · 2020-02-21

**Recommendation:** Accept
**Confidence:** 4

**Metareview:**

The reviewers appreciated the paper. They found the design study to be well conducted and insightful, but lacking to clear take-aways.

The main point to tackle while finalising the paper is to better distill the most interesting findings from the deployment. R2 suggests to put forward the most surprising ones and R3 notes the contributions s/he found most relevant.

The reviewers further encourage the authors to improve on the visuals related to the deployment :
 - A table or visual showing when and how often the different templates were used in a story (possibly per participant).
 - A figure to clarify the deployment.
 - Reduce the importance of the deployment tables

Both R2 and R1 suggest relevant papers on data storytelling and design research.

Finally while this may be more challenging, there is a wealth of literature on generalizabilty in Design Research (see R for a good entry point), considering this literature could help frame the contributions and its insights in a more situated and reflexive manner, rather than seeking generalisable findings.

---

### Decision · Program_Chairs · 2020-02-21

Accept